

# An Analysis of Swell and Bimodality Around the South and South-west Coastline of England

Daniel A. Thompson[1], Harshinie Karunarathna[2], Dominic E. Reeve[2]

[1]Jacobs Engineering Group, Burderop Park, Swindon, SN4 0QD, England, UK.

5  [2]College of Engineering, Swansea University, Bay Campus, Fabian Way, Swansea, SA1 8EN, Wales, UK.

*Correspondence to*: Dominic E. Reeve (d.e.reeve@swansea.ac.uk)





**Abstract**

This paper presents an analysis of wave recordings with particular attention to assessing bimodality of the incident wave energy spectra and the occurrence of swell along the south and south-west coasts of the United Kingdom, (UK). A procedure is developed to perform an intensive analysis of a new and large dataset of measured wave spectra. A storm during February 2014 is analysed in detail, highlighting the observed wave conditions leading up to and during the collapse of the sea wall at Dawlish, UK. The analysis reveals the prevalence of trapped-fetch conditions and long-period swell during the February 2014 storm. Bimodality and the presence of swell are compared at three locations along the south coast of the UK. Results highlight the increase in bimodality during the 2013/2014 storm period, especially at Dawlish. The analysis also provides evidence of bimodality and swell waves occurring far along the English Channel. Observed wave conditions at Dawlish are compared to the parametric limits of empirical formulae to estimate wave overtopping. There were numerous instances of peak wave periods or wave heights outside the limits of the formulae, showing that existing design formulae do not yet adequately account for the range of conditions experienced in coastal waters.

# 1 Introduction

Parametric empirical methods are commonly applied for the initial design and assessment of coastal defences and for hazard risk assessments. An example of this is the EurOtop wave overtopping manual, (Van der Meer et al., 2016), which provides empirical methods for estimating wave overtopping discharge. In the formulae proposed by EurOtop the wave period is characterised by the spectral wave period, $T_{m-1,0}$. The spectral wave period gives more weight to the longer wave periods in a spectrum and Van der Meer et al (2016) state it is suitable for multi-peaked spectra. The use of the peak wave period, $T_p$, is also used and applied to unimodal spectra. The simplification of a complicated sea state into a single parametric variable can be problematic and may mask the true nature of the sea state observed; potentially compromising the accuracy of estimates of wave overtopping and damage.

A bimodal distribution of wave periods has been observed to cause significant coastal damage and dramatic beach morphology change (Bradbury et al., 2007). A bimodal sea state occurs when a combination of swell and wind-waves are present within the same area of observation. When swell waves arrive in a region of locally generated wind-waves, the spectral shape of the sea state, which reveals the contribution of energy provided by the individual components of the sea state, shows two peak frequencies corresponding to the wave periods of the wind and swell components. Swell waves are defined as waves that have propagated away from the region of generation and are no longer affected by local winds. In the area of wind wave generation, high-frequency wave energy is both dissipated and transferred to lower frequencies (Reeve et al., 2004). The different wave frequencies cause waves to travel at different speeds. Dispersion occurs and the faster low-frequency waves form a swell and begin to propagate away from the area of generation. The formation of wind waves is



dependent on the wind speed, the width and length of the area over which the wind blows (fetch area), the duration of the wind, and the depth of water.

Under conditions in which the surface wave distribution is described by a bimodal frequency spectrum a single parametric
variable may not fully describe the observed sea state. By improving our understanding of swell and bimodal occurrences it may be possible to specify where a simplification of a sea state to a single parametric variable may or may not be suitable. This paper provides an analysis of bimodality and swell waves around the south and south-west UK with special interest on the 2013/2014 winter storms.

Swell waves are difficult to predict and wave impact can occur without warning as they are a product of distant storms, (Hawkes, 1999; Sibley and Cox, 2014). Swell waves have been known to cause significant overtopping of defences and beaches (Thompson et al., 2017). The UK coastline often experiences swell and bimodal events due to the presence of North Atlantic swell. In 1979 appreciable damage to sea defences and property was observed along the English Channel as a result of low-pressure weather systems and swell waves (Draper and Bownass, 1982). (Sibley and Cox, 2014) analysed seven
events where swell waves were present during coastal flooding. The swell waves were formed by North Atlantic storm systems which were directed towards the UK. Coastal flooding associated with swell waves was often found to correlate with tidal surges and within close proximity of fairly deep low-pressure centres. Such systems were observed during significant coastal damage along the UK coastline during the winter storms of 2013/2014 (Sibley et al., 2015). Parts of the Brazilian coast and the West African coast provide examples where significant swell can arrive unexpectedly at the coast
(Van der Meer et al., 2016). A history of global impacts arising from swell events is provided in Table 1 and is based on the work of  Palmer (2011) and Sibley and Cox (2014).

With burgeoning coastal monitoring programmes around the UK coastlines, there is now sufficient data of which suitable analysis can provide further insight into the occurrence of bimodality and swell waves. The aim of this paper is to analyse
wave spectra directly from wave buoy data during recent coastal flooding events and investigate the occurrence of bimodal seas and swell waves along the UK coastline and to provide an up-to-date picture of bimodality around the south and south-west of the UK. Firstly, the wave buoy data used in the analysis is introduced. The procedure for which bimodal seas are detected and swell percentage calculated is then explained. Time-series of observed wave conditions is presented during a significant storm whilst also highlighting the time and date of significant coastal damages. Locations are then chosen for
detailed analysis through the years and for comparisons against the parametric limits of overtopping formulae.



## 2 Wave Buoy Data

The Channel Coastal Observatory (CCO) provides data from the National Network of Regional Coastal Monitoring Programmes of England which consists of six Regional Monitoring Programmes. For each programme a lead local authority takes responsibility for funding applications, budget control, data collection, quality control, implementation of the programme and delivery to partners in the Regional Programme and Coastal Groups. The funding for the programmes is from central government via the Department of Environment, Food, and Rural Affairs (DEFRA), and is administered through the Environment Agency. The data available include LIDAR surveys, topographic surveys, aerial photography, tide gauge data, and nearshore wave buoy data.

Wave data from the wave buoy is available and downloadable in three hourly intervals as either a wave spectra or integrated wave parameters. The wave buoys are directional wave rider buoys supplied by Datawell BV. An archive is maintained of the wave parameters and spectra which is publicly available. The locations of the buoys are shown in Fig. 1 with location names provided in Table 2. The network of buoys was split into three sections, covering the south-west, south, and south-south-east which allows comparison of buoys in relatively similar wave conditions. For example, the buoys on the south-west are more likely to record swell conditions (due to openness to the Atlantic ocean). The moored wave buoy sites are all located in shallow water, typically around -10mCD.

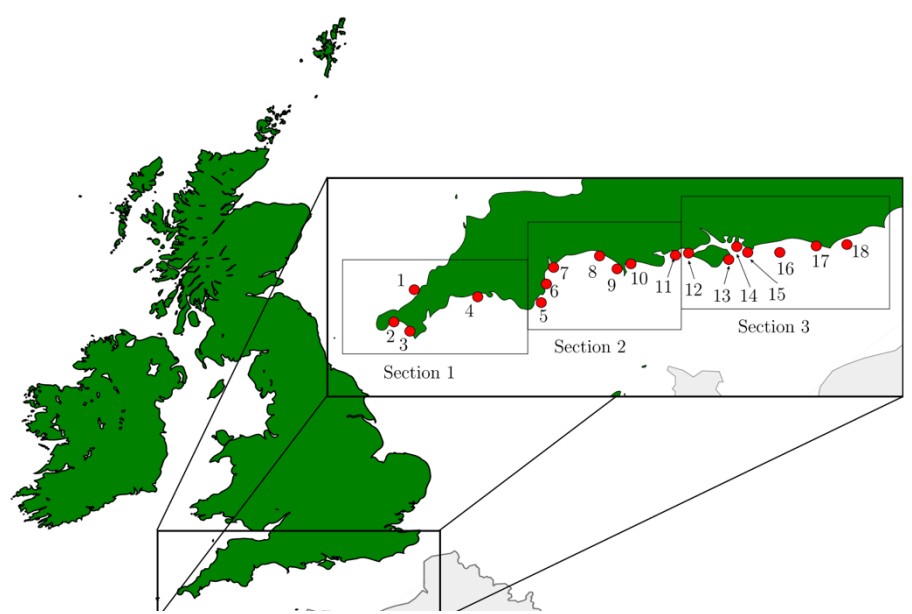

**Figure 1 - The locations of CCO buoys split into three sections. Location names provided in Table 2.**





For this analysis the raw spectra are used, unlike earlier studies, allowing flexibility in assessing the observed wave conditions. By using the raw spectra it is possible to derive the following parameters that will be used in the investigation $T_{m02}$, $T_p$, $H_s$, swell percentage ($P_{sc}$), energy content of the spectrum and whether the sea state was bimodal. In order to determine whether a bimodal sea state occurred, the spectra is partitioned into the swell component and wind component.

The approach used to determine the separation frequency, $f_s$, is provided in the next section.

## 2.1 Spectral Partitioning

The spectral wave buoy data provided by the CCO includes the energy associated with discrete frequency bands across the spectrum. The spectrum is discretised into bands of 0.005hz with cut-off frequencies of 0.025Hz and 0.58Hz, providing a total of 62 data points in the spectrum. The classification of a bimodal sea or the separation frequency between wind and

swell component is not provided in the data by the CCO. An automated process was thus developed to process the many thousands of spectra.

The method developed here is based on a filtering approach. The separation frequency (typically between 0.09 and 0.1Hz) is found and a filter is applied for the definition of appropriate bimodal conditions provided in (Bradbury et al., 2007). The

separation frequency is determined by automatically scrutinising the spectrum to find suitable peaks and troughs which fit the criteria of a bimodal spectra as defined below:

- The minimum total energy within the spectrum must provide an overall $H_s = 0.5$m.
- The smaller of the two spectral peaks must have a peak at least 1/3 of the energy density of the larger peak.
- The smaller of the two spectral peaks must have a power spectral density of $>0.4\text{m}^2/\text{Hz}$.

- The energy of the trough must be < half the energy of the smaller peak.

This algorithm was implemented in MATLAB code. These criteria ensure that when a bimodal event is identified there are two distinct peaks in the energy in each of the spectral components and that there is a reasonable amount of energy within each component. Comparison of the calculated separation frequency from the algorithm and that used on the CCO website is

given in Fig. 2 and show good agreement.





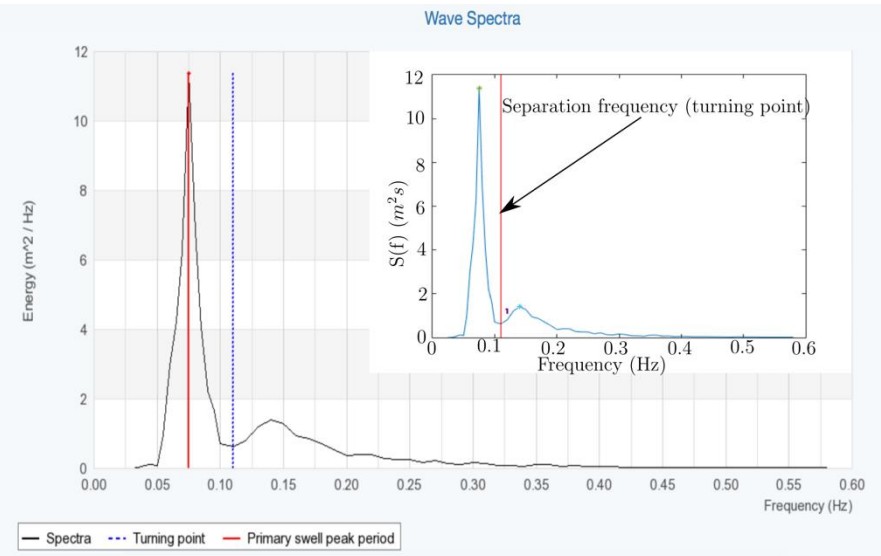

**Figure 2 - A comparison of computed separation frequency (inset) and that provided by the Channel Coast Observatory.**

The algorithm is an iterative procedure which first identifies the peaks and troughs in the spectrum. The highest peak and lowest trough are then marked. Tests are then performed based on the highest peak and lowest trough to find a suitable

secondary peak. Tests are performed until a suitable trough is found between the maximum and a secondary peak that lies within the defined separation frequency bracket (0.09 - 0.1Hz). Tests for each subsequent peak are performed to determine whether it meets the bimodal criteria via identifying any suitable troughs between the two peaks. Once a separation frequency is determined and two suitable peaks identified, wave parameters are then determined from the spectra. This includes the swell percentage, wind percentage, swell peak period, wind peak period, spectral period and wave height. In this

analysis swell percentage, $P_{sc}$, is the percentage of wave energy contained in the swell component. The energy in the swell component is the integral of the energy spectrum over frequencies from zero up to the defined separation frequency. The method does not classify multi-peak spectra (more than two peaks). If the algorithm is unable to choose a suitable separation frequency, the spectrum is flagged and the separation frequency chosen manually (by eye).

Example spectra and the computed separation frequencies at different locations are provided in Fig. 3. Examples of integrated wave parameters derived from the spectra shown in Fig. 3 are provided in Table 3.





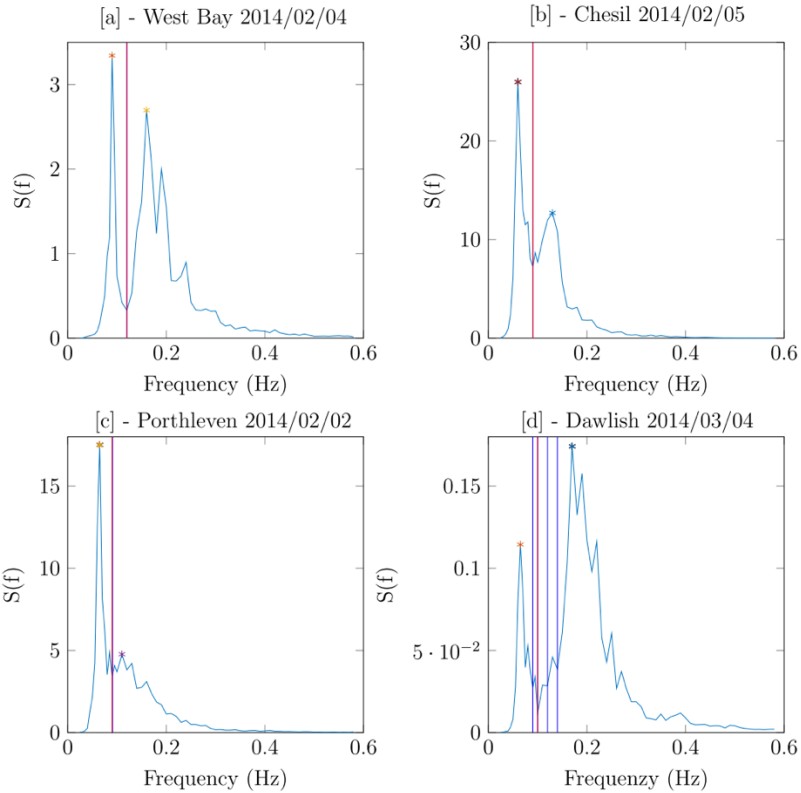

**Figure 3 - Example output spectra using the developed algorithm to determine the separation frequency of the wave buoy data. The solid blue lines in [d] represent troughs identified within the separation frequency bracket. The solid red line identifies the separation frequency.**

This approach was developed as it was not reliant on contemporaneous wind data, which was absent for several of the wave buoy locations. The method makes it possible to batch download wave buoy spectra from the CCO website and process thousands of individual wave spectra.

10 **3 Winter 2013/2014**

The UK lies on the east of the Atlantic Ocean, making it susceptible to deep Atlantic depressions which can be steered towards the UK by the upper tropospheric jet stream. The deep Atlantic depressions provide ample opportunity to generate energetic swell waves that travel towards the Atlantic facing shores of the UK. A summary of significant coastal flooding events are displayed in Table 1. More recent and significant coastal flooding events manifested themselves during the winter

15 period of 2013 and 2014 during the months of December (2013) to March (2014). Approximately 12 major storms occurred during the winter period. Of themselves the storms were not particularly extreme, however the clustering and persistence of the storms made the whole period very notable; making it the stormiest period of weather the UK had experienced for 20





years(Sibley and Cox, 2014). Sibley & Cox (2014) suggested that the storms most likely occurred as part of major perturbations to the Pacific and North Atlantic jet streams driven by persistent rainfall over Indonesia and the tropical West Pacific. The bulk of the ocean wave energy was displaced towards the south-west of Ireland and England as the tracks of storms during January and February had relatively low latitudes.

Total economic damages for England and Wales from the winter period of 2013 to 2014 were estimated to be between £1,000 million and £1,500 million, with a best estimate of £1,300 million. Chartteron et al., (2016) provide an in-depth breakdown on the costs and impacts of the winter 2013 and 2014 floods; together with a breakdown of the impacts on every sector; for example, transport, utilities, education, and agriculture. To assess whether damages varied by source of flooding,

an attempt was made to separate out the damages attributed to coastal impacts associated with the 2013 and 2014 floods. These enabled estimates to be made of the damages from coastal flooding; with the remaining damages being attributed to fluvial, groundwater and other sources such as pluvial. Due to insufficient detail in the flooding information, the separation assumed that the majority of the flood impacts experienced by the lead local flood authority areas during the winter 2013 to 2014 floods were caused by tidal surges or increased wave action, which is clearly a simplification. A summary of the

impacts and the coastal flood related damages in England and Wales is provided in Table 4 as well as their percentage contribution to the overall flood damage, (e.g. Coastal (63%), Fluvial/groundwater (37%)).

The purpose of providing Table 4 in this data analysis is to highlight the overall contribution coastal flooding played to the total damage caused by the winter storms. Table 4 demonstrates the wide impacts coastal flooding can have on society. This

includes the impact to wildlife sites, heritage sites and businesses. It highlights the importance of thorough analysis of what occurred during the winter storms in order to improve understanding of the events and help prevent similar or worse damage from occurring in the future.

Here, only one storm during the winter period of 2013/2014 is discussed in detail, the first storm in February which struck

the UK during the 3$^{rd}$ - 5$^{th}$ February. Comprehensive analysis of other storms during this period can be found in Thompson (2017). A detailed discussion of the meteorological conditions throughout the period is provided by Sibley et al. (2015). The track of the storms during this period contributed to extensive damage observed at many places along the west/south-west coast of the UK, and the formulation of significant swell wave due to trapped-fetch conditions, (Sibley et al., 2015). Trapped-fetch conditions develop when the main wave group induced by surface winds matches the speed of movement of

the low pressure system as a whole, (Sibley and Cox, 2014).

Results are presented in terms of a timeline of the wave conditions at each location. The timeline is split into three sections according to buoy location as presented in Fig. 1. The integrated wave parameters derived from the spectra used for this investigation are mean wave period, $T_{m02}$, peak wave period, $T_p$, significant wave height, $H_s$, observed swell percentage, $P_{sc}$




and energy content. Significant wave height and energy content are included to demonstrate the amount of energy within the sea state and to provide an indication of the arrival of storms. Observed swell percentage, the integral of the energy content below the separation frequency, provides a measure of how many low frequency waves are being observed. The parameter $P_{sc}$ is calculated irrespective of whether the conditions are bimodal or not. In the next section the observed conditions are

presented; followed by a discussion of the conditions in relation to damage and coastal flooding recorded at nearby coastal locations during the storm period.

## 4 Observed Wave Conditions

A rapidly deepening low pressure system moved across the Atlantic with an unusually low latitudinal track. Fig. 4 provides an example of the synoptic chart during this period. Gale-force southerly flow occurred in the English Channel on 3rd

February and record maximum water levels were measured at Plymouth with high tides and storm surges occurring at the same time. The dates included in the analysis are the 1st - 9th February. The timeline of key coastal damage events and estimated costs is provided in Table 5.

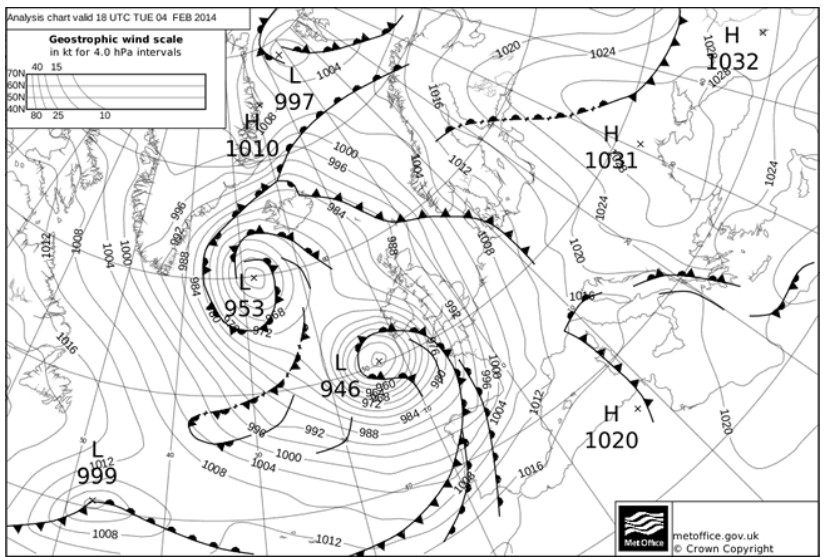

**Figure 4 - Synoptic chart at 18 UTC on 4th February 2014. (From UK Met. Office, 2015).**

In Section 1, $H_{m0}$ increases slightly with the arrival of the storm and $T_{m02}$ remains relatively stable, Fig. 5. All wave buoys experience high $P_{sc}$ (45%-85%). Compared to other sections a high $P_{sc}$ is found with relatively low $T_p$ and $T_{m02}$. A reason for this recorded increase in $P_{sc}$ is likely due to the way $P_{sc}$ is determined from the spectrum. The swell percentage is the integral of the energy in the spectrum below the separation frequency. A slight increase in $T_p$ is observed during the storm period

indicating the balance of energy in the spectrum shifts towards the lower frequencies. With increased wind sea conditions during a storm period, the separation frequency can be determined at a maximum of 0.1 Hz. Therefore, the swell percentage





may be large if energy content in the spectrum below the separation frequency is large, as would be indicated by the slight increase in observed peak wave period.

Unrealistically large wave heights are detected from the 5th February in Section 1. The $H_{m0}$ values indicate that the conditions after the 5th February may have increased to a point where the wave buoys were unable to accurately record data. Wave breaking will limit wave heights and so, derived $H_{m0}$ values greater than the water depth at the wave buoy, (see Table 2), must be deemed inaccurate. It is likely that the actual wave conditions were too large for the wave buoys to record measurements accurately, thus producing peculiar spectral shapes and unrealistic spectral parameters.

Porthleven experienced the greatest increase in energy content during the first day of the storm, 3rd February. The buoy at Perranporth detected the largest increase in energy content on the second day, 4th February. Perranporth is also the only location within Section 1 to encounter a continued long peak wave period, high swell percentage and long $T_{m02}$ into the second day of the storm, with a peak $T_p = 18$ s and $T_{m02} = 10$ s. Referring to Fig. 1 and the locations of the wave buoys, the Perranporth wave buoy is the only buoy located on the northern tip of the south-west. The location explains the observed differences in conditions and is the only location that detected a continued increase in wave height and period on the second day of the storm. The results for Section 1 suggest locations along the northern tip of the south-west are more exposed to severe and prolonged wave conditions generated by storms with a similar track as the one in this period. For all locations within Section 1, large peak periods are recorded on the 2nd February, a day when much wave overtopping and localised flooding occurred (see Table 5).

Extensive damage occurred in Section 2 at Dawlish during this storm period, where key railway protection infrastructure along the beach collapsed on the evening of 3rd and night of 4th February. A time estimate of the first indication of damage is shown in Fig. 6. As a major rail link between the south-west and the rest of England, the economic consequences of the damage were huge. In the days leading up to the collapse, high peak wave periods (9 s - 20 s) and swell components (35%-45%) are observed. During the first day of the storm (4th Feb), energy content, $H_{m0}$ and $T_{m02}$ are seen to gradually increase. However, all recorded wave parameters at Dawlish peak at the end of 5th February, the last day of the storm.

Sibley et al., (2015) suggested that it was the south-southeast wind generated waves in the English Channel in addition to the surge that was the initial cause of the damage. High $T_p$ is again observed during 2nd February just before the arrival of the storm. The arrival of the storm brings slightly larger waves to the Dawlish buoy and an increased $T_{m02}$ and it is potentially this combination, as well as the conditions suggested by (Sibley et al., 2015), of continuous longer-period waves reaching the sea wall and then the arrival of higher waves and longer $T_{m02}$ that caused the Dawlish sea wall to collapse. Chesil and Boscombe recorded high $T_p$ (20 s or more) and $P_{sc}$ (40% - 60%) leading up to the storm, during the storm $P_{sc}$ increases whereas $H_{m0}$ and $T_p$ only increase slightly. $T_{m02}$ is seen to increase at the end of 3rd and beginning of 4th February for all wave





buoys in Section 2. Very large increases in $H_{m0}$ and energy content are observed at the Chesil and Boscombe buoys after the main storm period (5[th] February) which coincided with significant shingle beach movement and warnings for localised flooding at Chesil.

5   No damage was recorded in Section 3, nevertheless conditions with a peak period of 15s prevailed at Milford and all wave buoys detected some level of swell, Fig. 7. A trough in $P_{sc}$ may be detected across all sections on the 3[rd] February. A noticeable time delay of this trough is observed from Section 1 through to Section 3 highlighting the movement of the storm (towards the east) and arrival of different wave conditions. A reason for the decrease in $P_{sc}$ could be associated with the arrival of the storm and strong local winds creating locally generated wind waves with high energy. The associated energy

10   content after this trough is also found to increase for all sections. Large values of $T_p$ may have been observed in Section 3 due to the unusually low latitudinal track of the low-pressure storm and the trapped fetch conditions generated.

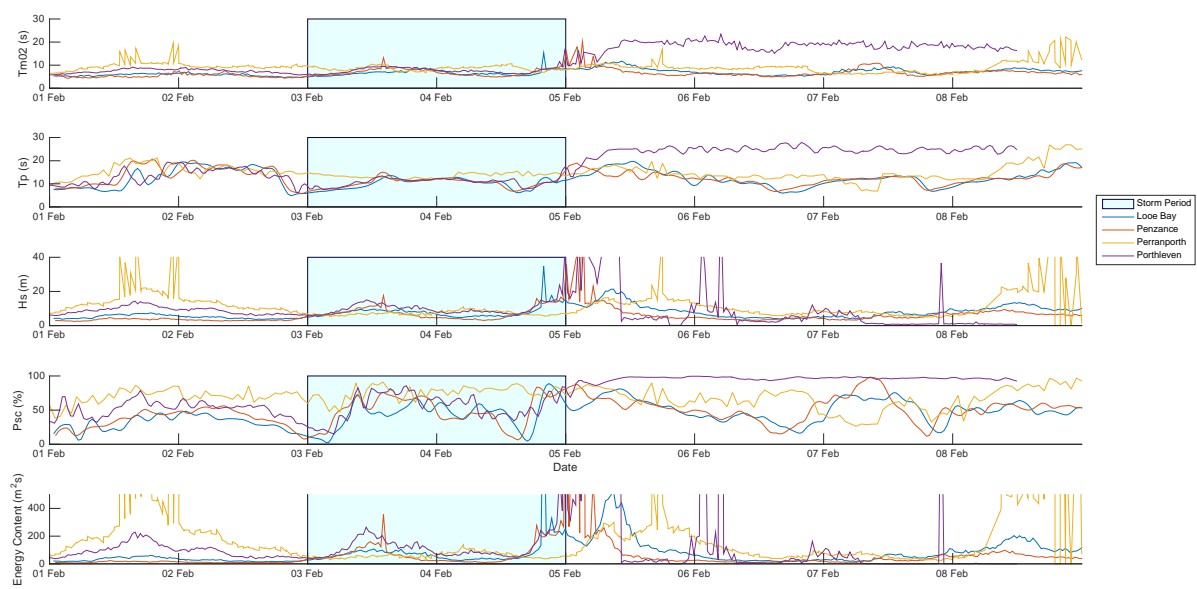

**Figure 5 - Timeline of the wave conditions derived from the wave buoy data during the 1st - 9th February - Section 1.**





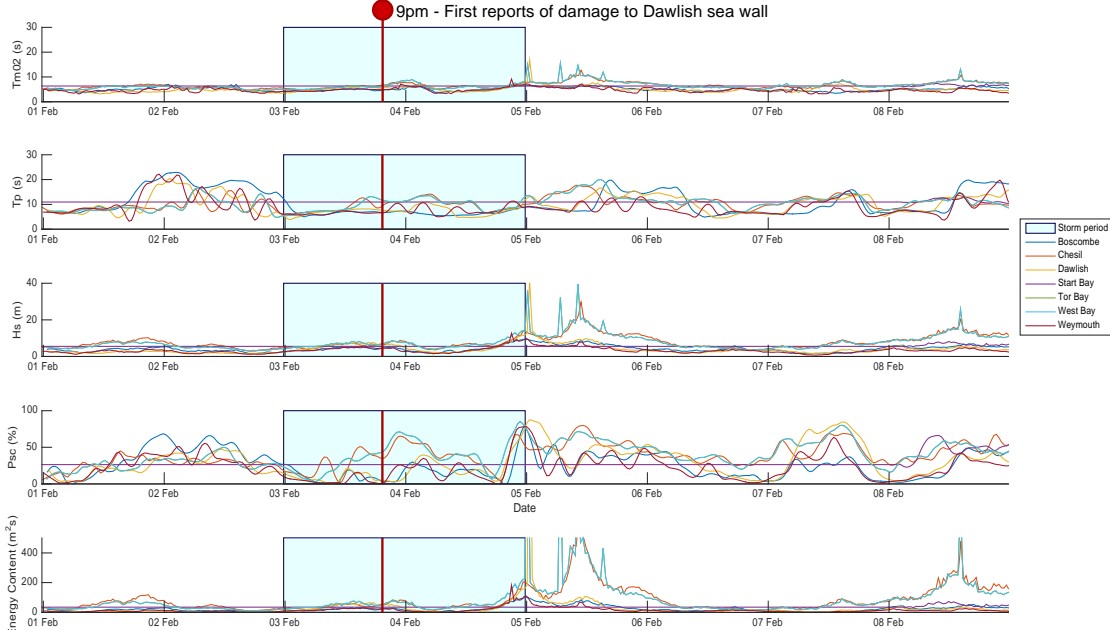

**Figure 6 - Timeline of the derived wave conditions from the observed wave buoy data during the 1st - 9th February - Section 2.**





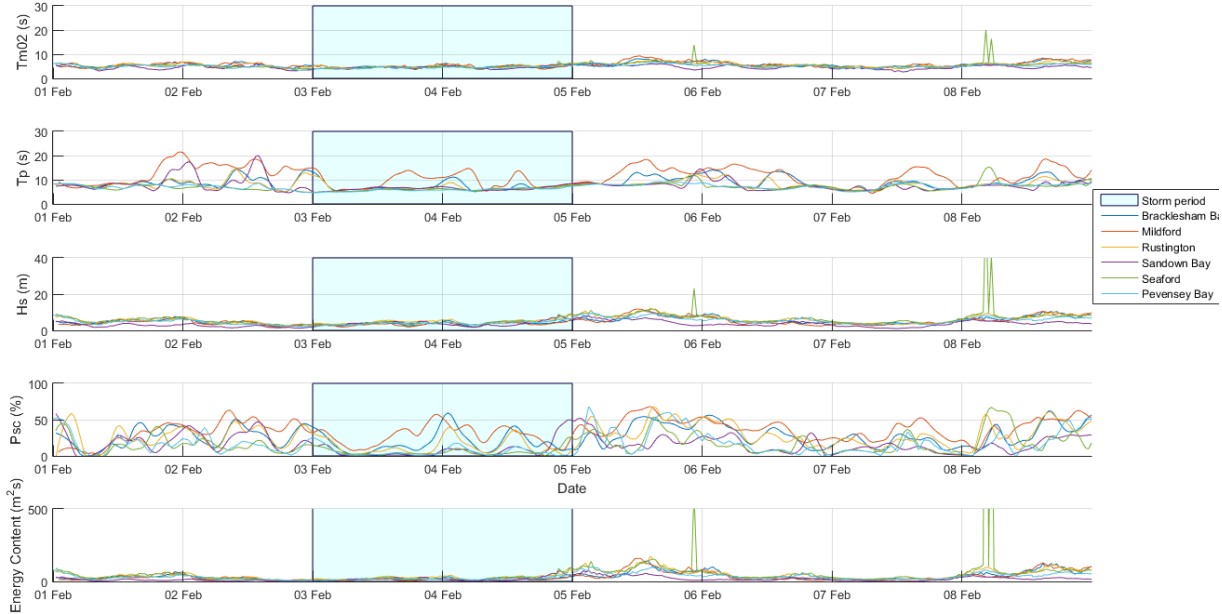

**Figure 7 - A timeline of the derived wave conditions from the observed wave buoy data during the 1st - 9th February - Section 3.**

One further point to note is that during the $1^{st}$ - $9^{th}$ February severe wave conditions continued after the main storm period. What makes this storm different to storms during the 2013/2014 winter is the unusually low latitude of the storm track. The movement of the storm clearly has an impact on the wave conditions that were observed during this period as the only clear increases that occur during this period are in $T_p$ and $P_{sc}$. All three sections detected the highest energy content after the storm period (on $5^{th}$ February) which could indicate the arrival of high energy long period swell slightly after the storm.

The observations during the winter storm period also provide evidence of propagation of long period swell in the English Channel, as also suggested by Bradbury (2007a). Figure 8 highlights the delay between the arrival of an increased $T_p$ across the three sections during period of $11^{th}$ February - $17^{th}$ February, the 'St Valentines Storm'. Peak wave periods firstly increase in Section 1, followed by several hours delay where an increase in $T_p$ is observed in Section 2, with peak waves reaching a slightly shorter period than found in Section 1. A few hours later, a peak in $T_p$ is found in Section 3 where the periods are shorter than both Section 1 and Section 2. As suggested by (Bradbury et al., 2007), knowledge of the arrival of swell would be advantageous and the observation of delay between sections could provide a means to forecast the arrival of swell.





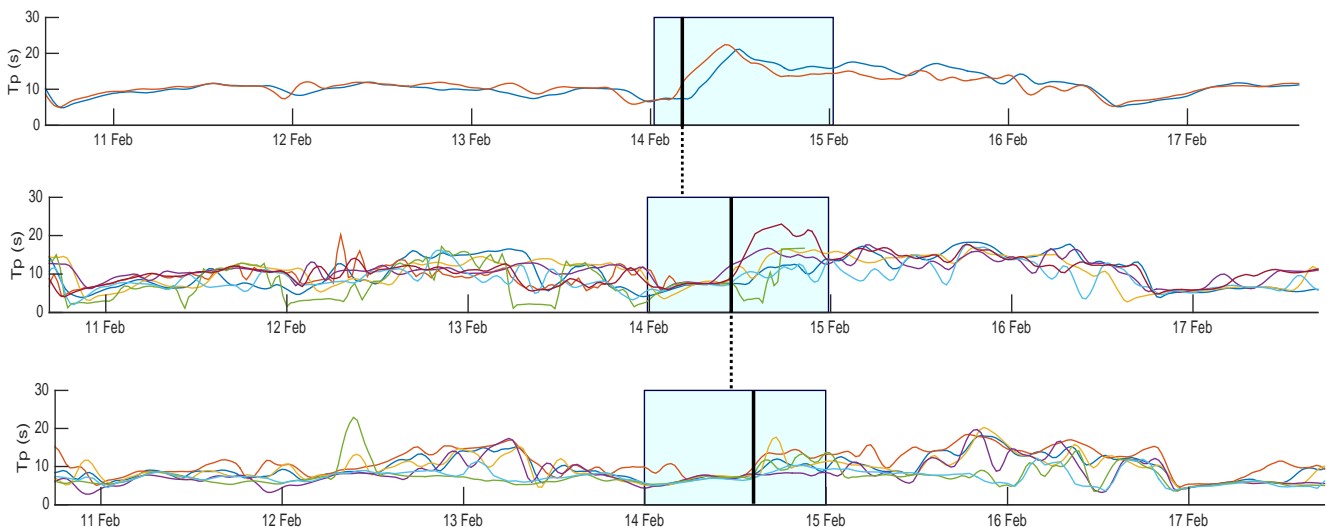

**Figure 8 - Demonstration of the delayed arrival of increased $T_p$ from Section 1 to Section 3 for Storm Period 3. The coloured lines represent the different locations within each section, Figs. 5 – 7 provide a key for this.**

The damage at Dawlish during this storm was considered a crisis to the southwest due to the cut in transport links and loss of

income for the local areas. What is not so immediately clear is whether the wave and tide conditions were correspondingly extraordinary. In the next section the conditions during February 2014 are put into a historical context by comparing them with observations from vicinal years.

## 5 Historical Comparison

Wave buoy recordings were collected for the months of January, February, March and December of each year from 2011 -

2017. Data was collected for one location within each of the Sections defined in Fig. 1 and Table 1.  Looe Bay was chosen from Section 1 as a location open to swell conditions propagating from the Atlantic. Dawlish was chosen from Section 2 due to the significant damage recorded during February 2014.) Rustington was chosen from Section 3 as a location potentially sheltered from swell conditions due to its location along the English Channel.

To make a comparison of the spatial variation within the observed wave conditions, the mean $P_{sc}$ of the month and the probability of bimodality are used. This mitigates the effects of variable sample size arising from measurement gaps due to damage and maintenance of wave buoys in comparisons between the months and datasets. Mean $P_{sc}$ will provide an idea of how often lower frequency waves were observed during the given months and the three locations. Higher mean $P_{sc}$ will indicate that during the month a high amount of energy in the low frequencies was encountered. The probability of a bimodal

sea occurrence, indicates how likely it was for the sea state to be a considered bimodal sea. If a higher probability value occurs, the occurrence of bimodality during that month was high. Figure 9 shows the results for three different locations.





**Figure 9 - Probability of bimodal occurrence and mean Psc for Looe Bay, Dawlish, and Rustington during winter months from 2011/2012 (11/12) to 2016/2017 (16/17).**





Bimodality during the winter period of 2013/2014 was frequent for both Dawlish and Rustington, which had the highest probability of a bimodal sea during winter months compared to other years, Fig. 9. For all three locations, the likelihood of bimodality was greatest during February of 2013/2014, the month during which the Dawlish sea wall collapsed. The probability of bi-modal occurrences is less than 10% in any winter month for all three locations. This is less than the 20%

value suggested by Bradbury et al. (2007) at Rustingon for the winter months of 2006 and 2007. Interestingly, Rustington encountered a similar probability of bimodality to Looe Bay, and in some cases, like 2015/2016, had a higher chance of bimodality Looe Bay. Rustington, located in Section 3 and in the south-east UK could be considered to be less likely to be subjected to swell as its location further east in the English Channel provides greater opportunity for swell waves to be refracted and diffracted towards the shore before penetrating further up the Channel. However, the analysis of the wave buoy

measurements shows that swell and bimodal seas are still quite commonly encountered here. It is clear, from the high probability of bimodality, that the swell generated during the winter storms of 2013/2014 propagated through the English Channel throughout December 2013 to March 201. Further, it was more likely for Rustington to experience a bimodal sea during February 2014, (8%), than at either Dawlish or Looe Bay. Unlike Dawlish, which is slightly sheltered from the Atlantic by a headland, Rustington is open to the English Channel, so any swell that propagates up the English Channel is

likely to be observed at Rustington, and will be important with regard to wave overtopping.

The probability of observing a bimodal sea at Dawlish is low, (see Fig. 9). However, the likelihood of a bimodal sea state occurring during the 2013 - 2014 winter storms increased materially. Again, as for mean $P_{sc}$, the peak in bimodality was observed during the month of the Dawlish sea wall collapse, February. The likelihood of a bimodal sea occurring during the

winter of 2013 - 2014 was much greater than in any other year between 2011-2016. This suggests that the strong Atlantic storms that occurred during 2013 - 2014, caused a significant increase in swell wave energy which, together with the trapped fetched conditions, created strongly bimodal wave conditions.

Looe Bay, located in Section 1, (Fig. 1), and in the south-west of UK, is susceptible to swell propagating from the Atlantic.

As seen in Fig. 9, the mean $P_{sc}$ and the probability of bimodal occurrence is higher at Looe Bay than Dawlish in Section 2 and Rustington in Section 3. The highest mean $P_{sc}$ is observed during the 2013 - 2014 winter storms with high values of $P_{sc}$ throughout December to March (27% -37%) and the highest mean value of $P_{sc}$ over all the years being in February 2014. The bimodality results at Looe are similar, where the highest probability of bimodal seas occurring was during the 2013 - 2014 winter storms and February 2014 having the highest probability (7%). This is in agreement with Table 5 where Looe

Bay is noted to have experienced localised coastal flooding. In December 2015, Looe Bay also experienced appreciable mean $P_{sc}$ together with high probability of bimodality.

From the above analysis we conclude that during the winter period of 2013/2014, the month of February was a highly unusual one in terms of bimodality and the presence of long period waves, and exhibited the greatest probability of a



bimodal sea and highest mean swell percentage at all three locations. Overall, Looe Bay experienced the highest mean swell percentage, followed by Rustington and then Dawlish. In terms of bimodality, Looe Bay and Rustington are similar, with the probability of bimodality being less. The results highlight how locations that are widely deemed to not experience much swell, Dawlish and Rustington, were faced with wave conditions containing a large swell component during February 2014.

The wave conditions during the winter storm period of 2013 - 2014 were unusual in comparison with other years, resulting in the damage that occurred along the coastlines. Given that coastal flood defences are designed on the basis of limiting wave overtopping, but most parametric formulae do not explicitly encompass bimodal or swell-dominated conditions, it is reasonable to consider whether the conditions experienced during the Winter 2013/14 period lie within the constraints of the
experiments used to derive the parametric design formulae.

## 6 Empirical Formulae Parametric Limit Comparison

In order to compare the observed wave conditions to current overtopping formulae parametric limits, the wave conditions were measured against the limits pertaining to the empirical formulae. An automated process was developed to compare the
input values of the overtopping formulae to the parametric limits, making it possible to compare several months of observations.

Due to the significant damage that occurred at Dawlish, the wave conditions at the Dawlish buoy were chosen to compare to the formulae parametric limits. The equations of Van der Meer et al. (2016) are derived from a collection of various
overtopping experiments referred to as the CLASH project. The parametric limits of CLASH were taken from Steendam et al. (2004) and the upper and lower limits of the offshore conditions are shown in Table 6. Comparisons have been made solely to EurOtop as it is currently the most widely used empirical method for estimating wave overtopping.

The wave height at the Dawlish buoy is plotted as a function of mean period, peak and spectral period in Figs. 10, 11, 12
respectively. The plots include only the values that lie outside of the limits defined in Table 6 during the winter months (December, January, February, March) from 2011 to 2016.





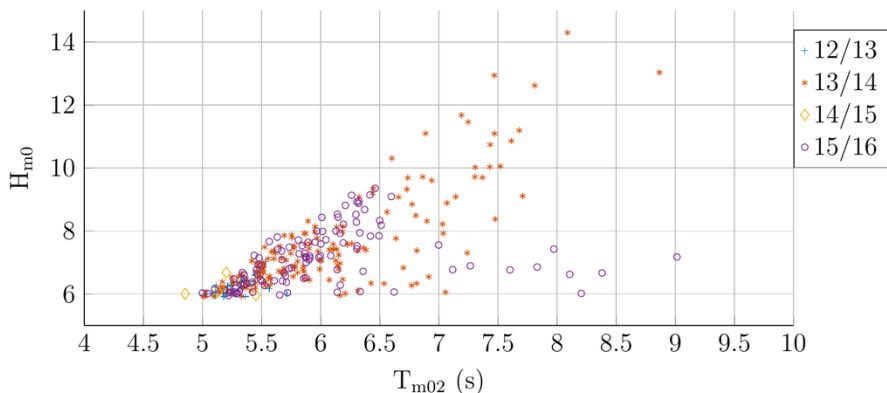

**Figure 10 - Comparison of observed H$_{m0}$ and T$_{m02}$ to parametric limits of overtopping formulae during the winter months from 2012 to 2016**

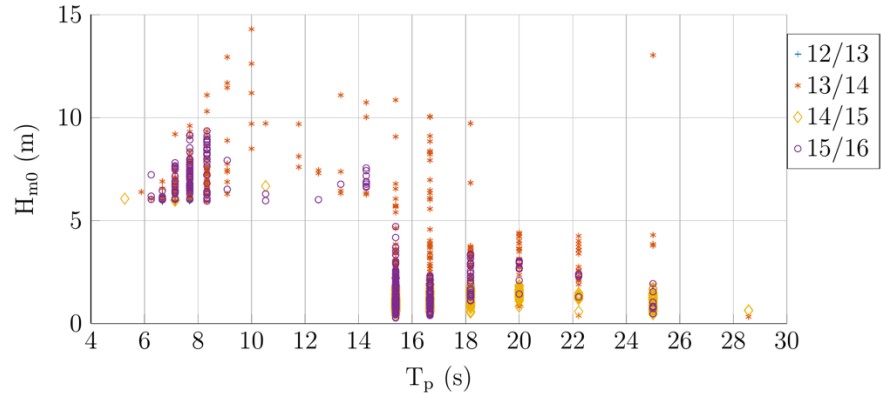

5   **Figure 11 - Comparison of observed H$_{m0}$ and T$_p$ to parametric limits of overtopping formulae during the winter months from 2012 to 2016**





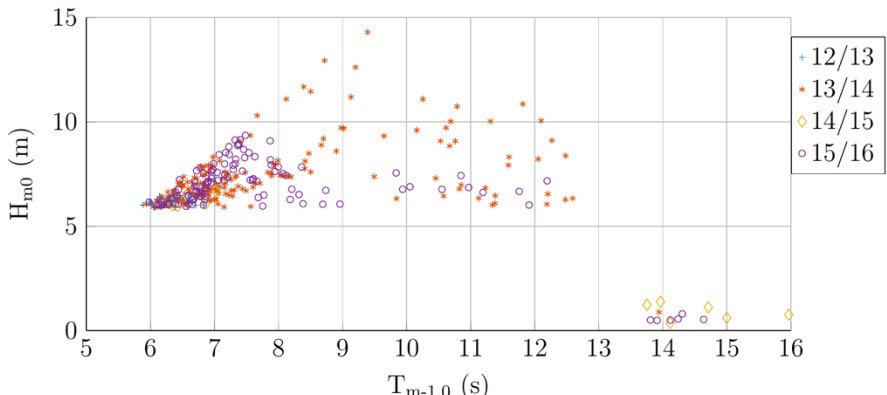

**Figure 12 - Comparison of observed $H_{m0}$ and $T_{m-1,0}$ to parametric limits of overtopping formulae during the winter months from 2012 to 2016**

5   It is clear that a considerable number of the wave conditions fall outside the parametric limits of the CLASH project, (10.5% of observed conditions during the 13/14 winter period). During the winter period of 13/14, long peak periods are observed with the largest wave heights when compared to other years. The Winter 14/15 period did not contain many cases of large wave heights, however there were many instances of long period waves as shown in Figs. 11 and 12.  Figure 11 also shows that wave conditions with peak wave period above 16 s, did not occur simultaneously with large wave heights.

From Figs. 10 - 12 it is possible to discern that the winter months of 13/14 and 15/16 contained materially larger waves with longer periods than the other years. For example, a large proportion of the observed spectral periods associated with a wave height greater than the upper limit are between $T_{m-1,0} = 6$ - 7.5 s each year; however during the years 13/14 and 15/16, similar wave heights are observed but also occur with a larger $T_{m-1,0}$. It is surmised that during years where coastal flooding and

15   erosion has been significant, there is a greater occurrence of observed conditions falling outside the parametric limits. At Dawlish wave height and peak wave period are the parameters that are more likely to fall outside the parametric limits.

The results imply that to predict wave overtopping fully over the period 2012-2016 would require the extrapolation of empirical methods beyond their limits of applicability. This is an important consideration when issuing flood warnings,

20   preparing flood risk assessments or developing initial design guidance. To fully replicate the conditions that fall outside the limits of the empirical formulae a scale physical model or computational model would be required.



## 7 Conclusions

The storms of the 2013 and 2014 winter period experienced by the UK created wave conditions that differed notably from those in recent years. They fashioned wave conditions that fell outside the limits of validity of empirical formulae for wave overtopping; containing a greater occurrence of bimodal seas and higher swell wave energy than other years. Damage and
localised flooding was ubiquitous along the south and southwest UK coastline. Our analysis has provided a strong indication that bimodal seas and swell waves play an important role in coastal flooding and damage, which is as yet not fully understood. The observed wave conditions indicate the occurrence of trapped-fetch wave conditions and that coastal flooding and damage correlates with bimodal sea conditions and mean swell percentage.

Our analysis highlights the importance of considering swell and bimodal seas, particularly when using design formulae that are couched in terms of integrated wave parameters. Design formulae, such as those used internationally for designing harbour breakwaters and coastal flood defences, were often developed using uni-modal wave conditions and their utility and veracity for conditions that do not confirm to this model is uncertain. Our analysis demonstrates that the formulae for wave overtopping do not cover the range of conditions that are experienced around the UK, a situation that is likely to be
replicated in other regions around the world. Continued observation and detailed analysis of wave conditions around our coasts is extremely important in this regard. With a changing climate and the possibility of changing wave conditions, understanding what these changes mean when applying empirical assumptions is crucial for coastal practitioners and researchers alike.

**Code Availability**

The procedure used in the paper was developed in MATLAB. The MATLAB code can be made available upon request.

**Data Availability**

The data studied in this paper can be accessed via the Channel Coastal Observatory: https://www.channelcoast.org/ .

**Author Contributions**

DT undertook the analysis and calculations. He also prepared the draft manuscript. HK provided technical guidance during the study and contributed to the editing of the manuscript. DR was DT's main supervisor and editor of the manuscript.

**Competing Interests**

The authors declare that they have no conflict of interest.

**Acknowledgements**




This work was supported by the Natural Environment Research Council as part of a PhD studentship (Grant No. EGF406).

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



**Table 1 - Timeline of recorded impacts due to swell waves.**

| Date | Location | Wave Characteristics | Observed Impacts |
|---|---|---|---|
| 1979 | English Channel (Draper and Bownass, 1982) | 9m wave height 20 - 25 s period | Extensive flooding at Chesil Beach, Hayling Island and other locations on the south coast. |
| 1983 | California and USA West Coast (Earle et al., 1984) | 12.9 m wave height 20 - 25 s peak period | Extensive erosion and damage along the USA West coast. |
| 1986 | English Channel (Nicholls and Webber, 1988) | 2 m wave height 16 s wave period | Overtopping of Chesil Beach and Hurst Beach. |
| 1987 | Maldives (Khan et al., 2002) | 2.5 m wave height 15 s period | Occurred during high tide, flooding, 30% of sand fill lost from reclaimed area. |
| 1991 | Nova Scotia to Florira (Maa and Wang, 1995) | 8 m wave height 20 s peak period | Waves affected the east coast of USA. |
| 2006 | English Channel (Sibley and Cox, 2014) | 2.4 - 3 m wave height and 11 - 13 s wave period | Coastal structure damage and flooded car parks in Devon |
| 2006 | Japan (Kashima, 2010) | 14.5 s wave period | Collapse of sea walls and inundation at Kuji Harbour. |
| 2008 | Carribbean Islands (Lefèvre, 2009) | 5 m wave height 18 s peak period | Exposed cities were flooded and serious damage occurred to roads. |
| 2008 | English Channel (Turton and Fenna, 2008) | 5 m wave height 18 - 20 s wave period | Coastal flooding along South-West UK coastline. |
| 2008 | Japan (Turton and Fenna, 2008) | 5 m wave height 18 - 20 s wave period | Coastal flooding along South-West UK coastline |
| 2008 | Japan (Yasuda et al., 2013) | 10 s wave period | Inundation and embankment breaches along Shimoniikawa coast. |
| 2012 | English Channel (Sibley and Cox, 2014) | 18 s+ wave period | Significant beach movement along South UK coastline |



| 2012 | English Channel (Sibley and Cox, 2014) | 14 - 20 s wave period | Beach loss along South UK coastline |
| --- | --- | --- | --- |
| 2013/2014 | South and West UK | 5+ m wave height 20 s+ wave period | A number of coastal flooding events and damages during winter period. |



**Table 2 - The location and reference of wave buoys from Fig. 1.**

| Section | Wave buoy number | Location | Approximate Water depth (m CD) |
|---|---|---|---|
| 1 | 1 | Perranporth | 14 |
|  | 2 | Penzance | 10 |
|  | 3 | Porthleven | 15 |
|  | 4 | Looe Bay | 10 |
| 2 | 5 | Start Bay | 10 |
|  | 6 | Tor Bay | 11 |
|  | 7 | Dawlish | 11 |
|  | 8 | West Bay | 10 |
|  | 9 | Chesil | 12 |
|  | 10 | Weymouth | 10.6 |
|  | 11 | Boscombe | 10.4 |
| 3 | 12 | Milford | 10 |
|  | 13 | Sandown Bay | 10.7 |
|  | 14 | Hayling Island | 10 |
|  | 15 | Bracklesham Bay | 10.4 |
|  | 16 | Rustington | 9.9 |
|  | 17 | Seaford | 11 |
|  | 18 | Pevensey Bay | 9.8 |





**Table 3 - Example spectral wave parameters derived from the spectra in Fig. 3.**

| Location | $H_{m0}$ (m) | $T_{m-1,0}$ (s) | $T_{p,wind}$ (s) | $T_{p,swell}$ (s) | Swell Percentage (%) |
|---|---|---|---|---|---|
| Dawlish | 1.34 | 7.46 | 5.88 | 15.68 | 24.69 |
| Porthleven | 9.46 | 11.90 | 9.09 | 15.38 | 56.26 |
| West Bay | 5.22 | 7.39 | 6.25 | 11.11 | 33.16 |
| Chesil | 13.77 | 12.12 | 7.69 | 16.67 | 54.29 |



**Table 4 - Coastal flooding impacts and the associated damage costs as detailed in** (Chartteron et al., 2016)**.**

| Impacts on: | Best estimate of damage (£million) | Percentage of coastal flooding contribution to overall flood damage (%) |
|---|---|---|
| Residential properties | 130 | 40 |
| Businesses | 170 | 63 |
| Temporary accommodation | 20 | 40 |
| Motor vehicles, boats and caravans | 15 | 40 |
| Local authorities and local government | 37 | 65 |
| Flood risk infrastructure | 110 | 75 |
| Utilities: water | 0.38 | 1 |
| Transport: road | 70 | 39 |
| Transport: rail | 17 | 74 |
| Transport: ports | 1.8 | 100 |
| Public health and welfare | 9.8 | 40 |
| Education | 0.92 | 56 |
| Agriculture | 0.21 | 1 |
| Wildlife sites | 2.3 | 95 |
| Heritage sites | 5.9 | 79 |
| Tourism and recreation | 2 | 56 |



**Table 5 - Locations of observed damage with estimated cost of the damages during the storm period. Details collected from local newspapers.**

| Date | Location | Observed Damage | Permanent estimate cost |
|------|----------|-----------------|-------------------------|
| 2/2/14 | Across Devon | Collapse of walkways, lifting slabs, damage to beach access and railings | N/a |
| | Looe Bay | Localised flooding | N/a |
| | Perranporth | Localised flooding | N/a |
| | Bude | Localised flooding | N/a |
| | Porthreath | Localised flooding | N/a |
| | Trevone, Padstow | Beach wall collapse | N/a |
| | Newquay | Sea front damaged and road collapse | |
| 4/2/14 | Dawlish | Sea wall collapse | £35 million |
| | Newton Abbott to Paignton | Sea wall collapse – rail closures | N/a |
| | Newton Abbott to Plymouth | Sea wall collapse – rail closures | N/a |

**Table 6 - Parametric limits of taken from the CLASH project (Steendam et al., 2004).**

| Parameter | Lower limit | Upper limit |
|-----------|-------------|-------------|
| $H_{m0}$ | 0.003 | 5.92 |
| $T_p$ | 0.545 | 15 |
| $T_m$ | 0.454 | 12.5 |
| $T_{m-1,0}$ | 0.495 | 13.6 |