# Peer review of "An Analysis of Swell and Bimodality Around the South and Southwest Coastline of England"

_Natural Hazards and Earth System Sciences, 2018_

## Short Comment (SC1) · 11 Jun 2018

A very interesting paper with some important implications for coastal engineering. I have a few minor comments:

1. The wave buoy spectra are available from the CCO at 30 minute intervals, not 3-hourly

2. Please note that your Figure 2 shows the location only of the buoys you are using for this research (the text implies that these are the only buoys, but there are similar buoy locations around the whole English coast)

3. The reference which gave the criteria for defining a bimodal spectrum wasn't Bradbury et al 2007, but Mason et al 2008. Full reference is: Mason T, Bradbury A, Poate, T

& Newman R (2008). Nearshore wave climate of the English Channel - evidence for bi-modal seas. Proceedings of the 31st International Conference on Coastal Engineering, American Society of Civil Engineers, 605-616

---

## Referee Comment (RC1) · Anonymous Referee #1 · 1 Aug 2018

The aim of this work is to analyze how the bimodality of the waves along the UK coastline can influence on coastal flooding extreme events with significant damages and highlight the importance of re-defining the empirical formulation for the design of marine infrastructures (in this study, overtopping) to take it into account. Instrumental records from buoys along the south UK coast are analyzed to identify the probability of bimodality and the percentage of the swell components. Most of the analysis is focused in the exceptional winter 2013/2014 where severe coastal events with important damage occurred, especially in Dawlish location.

Although the historical characterization of the bimodality and swell percentage for 2011-2017 winters reveals exceptional conditions at Dawlish, this information is not enough to conclude that these conditions determinate the severe damages during
the winter 2013/2014. There are some other aspects that should be considered: the coastal flooding is produced by the combination of different marine dynamics (also storm surge and astronomical tide), the previous configuration of the beach or the coastal protection due to the sequence of storms during that winter (the damages at the beginning of February could be the result of accumulated ones). Therefore, the magnitude of the damage might no be proportional to the magnitude of the coastal flooding event.

It could be interesting to apply the overtopping formulation (or a total water index) and compare the evolution of these time series during 1st-9th of February 2014 with the wave conditions and with the sea level (storm surge and astronomical tide) to analyze which is the percentage of influence of each component. On the other hand, to be sure that bimodality determinates the magnitude of the coastal flooding event and the magnitude of the damage, this analysis should be done for another storm events in that area.

It is relevant to demonstrate that coastal engineering formulations should be updated because the characterization of a sea state as single parametric variable is too simplistic. However, identifying the wave conditions outside the limits of the parametric formulations does not prove how this affects the results of the overtopping. It might be interesting to compare the overtopping magnitude for different extreme events if an underestimation (or overestimation) is obtained due to wave bimodality and/or a significant swell component. Besides, it is not clear if wave conditions outside the limits of applicability of empirical formulation correspond with bimodality conditions.

---

## Referee Comment (RC2) · Anonymous Referee #2 · 15 Aug 2018

General comments:

This paper investigates several topics related to important engineering applications, such as wave overtopping combined with the probability of a higher occurrence of spectrum bimodality. The research concentrates on one specific climatic event (Winter 2013/2014) and one specific place of failure (Dawlish). The use of simplistic formulations in coastal structure designs is a vital research topic, even though computational capacities have become more powerful in recent years. Aspects such as the increased complexity of the directional spectrum and how this complex wave behavior must be included in new engineering equations is of upmost importance. However, several important aspects must be included in the analysis to help understand the physical problems and the future inclusion of these aspects in new formulations. Firstly, I agree with

most of the comments mentioned by the reviewer that highlight the need to understand and include other marine dynamics such as storm surges, astronomical tides, beach configuration and even sea level rise (see Dawson et al, 2016). I also agree that it is important to perform the analysis including not only the wave bimodality but also other wave components in the directional spectrum (e.g. the significant swell component mentioned by the reviewer or even other complex systems). To investigate this clearly, it is necessary to perform an analysis for different extreme events that includes the conditions mentioned above combined with a more complex understanding of atmospheric (spatial wind patterns, synoptic charts through the storm track) and wave dynamics in the research area (spatial behavior of main wave parameters such as Significant wave height for sea and swell systems, peak periods, peak directions, directional spectrum, among others). Bimodality of the spectrum has a strong connection with local and far from climate conditions (local sea and far from swell systems), strong influence in the evolution of the spectrum of the no linear wave-wave interactions was observed. Local wind direction and main peak directions and frequencies of the directional spectrum might be very important in the analysis of bimodality. But the research does not consider any kind of directional analysis of the ocean waves reaching the research area. Despite the well-known difficulties obtaining the directional spectrum, I consider that it is important to try to include this aspect. As we know directionality (even in a simple form – using angle of wave attack) is very important for most of the empirical formulations of wave overtopping. It is known that the wave pressure and overtopping for obliquely incident waves are smaller than those for normally incident waves. All the comments mentioned above might be useful to find the main characteristics of wave bimodality (when compare with other wave conditions) in combination with other marine dynamics that increase the probability of coastal flooding and will be a strong scientific contribution. Some minor comments and some related to the topics mentioned above are shown below:

Specific comments through the manuscript:

1) Line 24: Please include other scientific references confirming this behavior. Include the paper called Computational Investigation of the Effects of Bimodal Seas on Wave Overtopping by Thompson et al. (2016). 2) Wave buoy data section. Comment: Please consider including a brief description of the study area, and please include a description of the general climate during the year. What about the Met Office's network of Marine Automatic Weather Stations (MAWS) consisting of 11 moored buoys? 3) Figure 1. Please include more details on this figure: coordinates, names, a map with a general location, etc. A map showing the continental shelf (including bathymetry) might be useful given the importance of this area on wave process. Please use a more detailed title that includes the research area. Please describe more details about the spectrum construction. Some instrumental buoys present several drawbacks related with these estimations. Please include the names of the buoys employed for the main analysis in Looe Bay (Section 1), Dawlish (Section 2) and Rustington (Section 3). 4) Partitioning method. Given the possible differences between the partitioning methods, please explain why you not did use other partitioning methods. 5) In the manuscript, the following asseveration is included: The method does not classify multi-peak spectra (more than two peaks). If the algorithm is unable to choose a suitable separation frequency, the spectrum is flagged and the separation frequency chosen manually (by eye). Given these drawbacks why were other methods not tested? 6) Figure 3. Please include the energy spectrum units (m2/Hz) 7) Title 3 Winter 2013/2014. Please consider having a more clear title for this section, e.g. Description of climatology during the 2013/2014 winter season. 8) In the next text: The purpose of providing Table 4 in this data analysis is to highlight the overall contribution coastal flooding played to the total damage caused by the winter storms. Comment: This is the main reason why the climate must be explained briefly in the study area section proposed. 9) In the text: Significant wave height and energy content. Comment: Please explain the concept employed for energy content. The connection between Psc and energy content is not clear. Please explain the exact definition of energy content. 10) In general: Please include in the Figures all the places mentioned throughout the paper. 11) Figure 4. Please improve the quality

of this figure. Try to add the coordinates in latitude and longitude. Please try to edit the original Synoptic chart to increase the quality of the coastline using another color or line style. Please add the bottom boarder line of the figure. Is possible to add the evolution of the storm toward the research area for different dates? See the following link as an example. http://www.mwis.org.uk/synoptic-charts. 12) In the text: In Section 1, Hm0 increases slightly with the arrival of the storm. Comment: Please clarify this analysis. The exact date of this analysis is not clear. A joint analysis of storm track information and wave parameters must be performed clearly. 13) In the text: unrealistically large wave heights are detected from the 5th February in Section 1. Comment: Please discuss the reason why this behavior (Unrealistically large wave heights) is not observed in the Perranporth location. I suggest including information about the track of the storm and some directional information about the ocean waves. The direction of the swell pattern traveling toward the research area can be very important for wave evolution through time. The evolution of storm direction can help to explain this behavior. 14) In the text: Porthleven experienced the greatest increase in energy content during the first day of the storm, 3rd February. The buoy at Perranporth detected the largest increase in energy content on the second day, 4th February. Comment: with a low value when compared to Porthleven. Try to discuss the relationship with the buoy location. 15) In the text: Perranporth wave buoy is the only buoy located on the northern tip of the south-west. The location explains the observed differences in conditions and is the only location that detected a continued increase in wave height and period on the second day of the storm. The results for Section 1 suggest locations along the northern tip of the south-west coast are more exposed to severe and prolonged wave conditions generated by storms with a track similar to the one in this period. Comment: This is one of the most important topics. For that reason, the quality of the discussion could be increased if a pattern of the spatial wave and storm evolution could be included in the paper. This would help to understand wave transformation toward the research area (other physical processes can be explained better). 16) In the text: Extensive damage occurred in Section 2 at Dawlish during this storm period, where key railway protection infrastructure along the beach collapsed on the evening of 3 rd and night of 4th February. Comment: Is it possible to include some damage photographs? During this event strong winds were observed, so please consider including some wind patterns (scatterometers could be a good option). 17) In the text: In the days leading up to the collapse, high peak wave periods (9 s - 20 s) and swell components (35%-25 45%) are observed. Comment: Please consider putting "were". 18) In the text: Sibley et al., (2015) suggested that it was the south-southeast wind generated waves in the English Channel. Comment: Please consider putting the name of the channel in Figure 1. 19) In the text: A reason for the decrease in Psc could be associated with the arrival of the storm and strong local winds creating locally generated wind waves with high energy. Comment: If the local wind waves are very important, please consider including some information about local wind (magnitude, direction, spatial maps, among others). 20) Figures 5, 6 and 7: Please increase the quality of these Figures. As the y axis is too narrow, it is difficult to read the values. Some values look incomplete (cut off at the top of the figure). Discuss the evolution using exact values. 21) Have you considered showing the evolution of the unidirectional spectrum by showing the evolution of the sea and swell simultaneously through time?. 22) Historical comparison: Explain why these months were selected for the analysis. This topic supports the importance of a study area section when the main climatological aspects in the research area must be discussed. A discussion of the annual cycle or stormy season during the year can be helpful. The estimation of the Psc during the month is not clear, please explain better. 23) In the text: Bimodality during the winter period of 2013/2014 was frequent for both Dawlish and Rustington. Comment: Looe bay has similar behavior; please consider including it in the analysis. 24) In the text: Rustington, located in Section 3 and in the south-east UK could be considered to be less likely to be subjected to swell as its location further east in the English Channel provides greater opportunity for swell waves to be refracted and diffracted towards the shore before penetrating further up the Channel. Comment: This confirms the importance of using an ocean wave model like SWAN or WWIII to model the spatial behavior of the wind waves during these dates.

[Figure]

25) In the text: Bimodality during the winter period of 2013/2014 was frequent for both Dawlish and Rustington: Comment: Please try to connect the fact that for Dawlish the highest percentage of bimodality was observed during February 2013/2014 when compared with the other years and the fact that the sea wall collapsed during this month. A technical discussion could be very useful. Why does bimodality increase the failure probability? Looe bay has similar behavior; please consider including it in the analysis. The connection between bimodality and failure probability is not clear. Is there other evidence of bimodality and the failure of other structures? 26) In the text: As seen in Fig. 9, the mean Psc and the probability of bimodal occurrence is higher at Looe Bay than Dawlish in Section 2 and Rustington in Section 3. The highest mean Psc is observed during the 2013 - 2014 winter storms with high values of Psc throughout December to March (27% -37%) and the highest mean value of Psc over all the years being in February 2014. Comment: This conclusion is not totally clear when comparing Looe Bay and Rustington; please verify.

References: Dawson, D., Shaw, J and Gehrels, W., 2016. Sea level rise impacts on transport infrastructure: The notorious case of the coastal railway line at Dawlish, England. Journal of Transport Geography 51 (2016) 97–109.

Please also note the supplement to this comment:
https://www.nat-hazards-earth-syst-sci-discuss.net/nhess-2018-117/nhess-2018-117-RC2-supplement.pdf

―――――――――――――

---

## Author Comment (AC1) · 20 Oct 2018

| Page | Reference | Comment | Action |
|---|---|---|---|
| | | **Replies to Short Comment** | |
| | | The wave buoy spectra are available from the CCO at 30 minute intervals, not 3-hourly | The text will be amended accordingly. |
| | | Please note that your Figure 2 shows the location only of the buoys you are using for this research (the text implies that these are the only buoys, but there are similar buoy locations around the whole English coast) | The supporting text will be amended to clarify this. |
| | | The reference which gave the criteria for defining a bimodal spectrum wasn't Bradbury et al 2007, but Mason et al 2008. Full reference is: Mason T, Bradbury A, Poate, T & Newman R (2008). Nearshore wave climate of the English Channel - evidence for bimodal seas. Proceedings of the 31st International Conference on Coastal Engineering, American Society of Civil Engineers, 605-616 | The reference will be amended accordingly. |
| | | | |
| | | | |

| | | **Reviewer 1** | |
|---|---|---|---|
| General | General | This information is not enough to conclude that these conditions determinate the severe damages during the winter 2013/2014 | Thank you for the comment. We agree and do not claim this in our conclusions. We argue that the bimodal conditions and their unusual severity which take the conditions outside the realm of empirical formulae used to design seawalls are a contributing factor to the failure of the Dawlish sea wall. This is an aspect of wave overtopping has not been analysed in this manner before. |
| General | General | The coastal flooding is produced by the combination of different marine dynamics (also storm surge and astronomical tide) | Yes. The text in the Introduction will be amended to clarify. |
| | | The previous configuration of the beach or the coastal protection due to the sequence of storms during that winter (the damages at the beginning of February could be the result of accumulated ones) | Yes. The text in the Introduction will be amended to clarify. |
| | | It could be interesting to apply the overtopping formulation (or a total water index) and compare the evolution of these time series during 1st-9th of February 2014 with the wave conditions and with the sea level (storm surge and astronomical tide) to analyze which is the percentage of influence of each component. | While it would be possible to perform an analysis 'with and without' surge, wave overtopping is a strongly nonlinear function of both water level and wave conditions. While a simple linear deconstruction is possible, any interpretation would be uncertain. That being said, the empirical analysis performed in Section 6 covers this to some extent, although the problem of how to make predictions when conditions lie outside the limits of validity of the empirical formulae remains. |

| | | | On the other hand, to be sure that bimodality determinates the magnitude of the coastal flooding event and the magnitude of the damage, this analysis should be done for another storm events in that area. | Our results suggest that bimodality is a contributory factor. An exact analysis of its influence will require a suitable research programme.  We have analysed other storms, which indicate similar trends to those shown in the paper. These additional analyses may be found in the first author's PhD thesis, listed in the References, and have not been included here for the sake of brevity and clarity. |
|---|---|---|---|---|
| | | | identifying the wave conditions outside the limits of the parametric formulations does not prove how this affects the results of the overtopping. | Yes, this is true. However, the empirical formula has a strong positive dependence between overtopping and the wave variables (height and period). This, and physical consideration of the energy contained in waves, would lead one to anticipate that waves outside the limits with larger heights and larger periods would lead to greater overtopping; but this has not yet been proven in laboratory experiments. |
| | | | | Further, the paper was not aimed at answering this question but rather to highlight the extreme wave conditions observed during this period compared to other years and to raise the fact that these recent storm conditions were outside the limits for empirical formulae that are used to design seawalls, both in terms of their wave height-period combination and that there spectra were bimodal rather than unimodal. The analysis highlights that wave conditions outside current parametric limits are possible in nature and have occurred. |

| | | It might be interesting to compare the overtopping magnitude for different extreme events if an underestimation (or overestimation) is obtained due to wave bimodality and/or a significant swell component. | We agree and this has been studied in detail in the first author's PhD thesis. Including this here would take the discussion in the paper well beyond its stated scope. |
| | | Besides, it is not clear if wave conditions outside the limits of applicability of empirical formulation correspond with bimodality conditions. | Yes, it is true that being outside the limits of applicability of the empirical formulation does not necessarily imply bimodality. However, for the storm conditions discussed in this paper the conditions were all bimodal. |

| No. | Location | | Action |
|---|---|---|---|
| | | **Reviewer 2 (including supplement)** | |
| 1 | Line 24 | Please include other scientific references confirming this behaviour. Include the paper called Computational Investigation of the Effects of Bimodal Seas on Wave Overtopping by Thompson et al. (2016). | Other references will be added |
| 2 | Wave buoy data section | Please consider including a brief description of the study area, and please include a description of the general climate during the year. What about the Met Office's network of Marine Automatic Weather Stations (MAWS) consisting of 11 moored buoys? | We will include a brief description of the study area and its characteristics. The MAWS is a good source of wave information but the buoys are located in deep water whereas the CCO wave buoys are all in shallow water, which is more relevant for the topic of this paper. |

| | | detailed title that includes the research area. Please describe more details about the spectrum construction. Some instrumental buoys present several drawbacks related with these estimations. Please include the names of the buoys employed for the main analysis in Looe Bay (Section 1), Dawlish (Section 2) and Rustington (Section 3). | The requested buoy details will be included. Details of spectrum construction are available on the CCO website. We can provide the link or include brief details in the revised manuscript. |
|---|---|---|---|
| 4 | Partitioning method | Given the possible differences between the partitioning methods, please explain why you not did use other partitioning methods. | There are many methods proposed in the literature. To be consistent with the coastal engineering literature we chose the method used by the CCO. We can include this explanation in the revised manuscript. |
| 5 | Text: "The method does not classify multi-peak spectra (more than two peaks). If the algorithm is unable to choose a suitable separation frequency, the spectrum is flagged and the separation frequency chosen manually (by eye). " | Given these drawbacks why were other methods not tested? | There are two reasons:

1) At present most coastal engineering sea defence design formulae for random wave conditions use or assume a unimodal spectrum as default. Considering a bimodal spectrum is a substantial conceptual shift. Proposing a multi-modal spectrum would be considered an unnecessary over-complication by many practitioners;
2) We are interested primarily in the binary categorisation of the wave conditions into 'wind sea' and 'swell'. |

| | | | We have chosen to use a similar method to that proposed by the CCO, which is widely recognised within the coastal science and engineering communities. This provides consistency in the analysis of the raw data. |
|---|---|---|---|
| 6 | Figure 3 | . Please include the energy spectrum units (m2/Hz) | This will be added to the figure |
| 7 | Title 3 Winter 2013/2014 | Please consider having a more clear title for this section, e.g. Description of climatology during the 2013/2014 winter season. | We will change the title to: " Description of climatology during the 2013/2014 winter season." |
| 8 | Text: The purpose of providing Table 4 in this data analysis is to highlight the overall contribution coastal flooding played to the total damage caused by the winter storms. | This is the main reason why the climate must be explained briefly in the study area section proposed. | A brief description will be included as per comment (2) above. |

| 9 | text: Significant wave height and energy content. | Please explain the concept employed for energy content. The connection between Psc and energy content is not clear. Please explain the exact definition of energy content. | Energy content is the integral of the energy spectrum over frequency. We will add the definition of 'energy content' and expand the explanation of Psc. |
|---|---|---|---|
| 10 | General | Please include in the Figures all the places mentioned throughout

the paper. | We will add these in the revised manuscript. |
| 11 | Figure 4 | Please improve the quality of this figure. Try to add the coordinates in latitude and longitude. Please try to edit the original Synoptic chart to increase the quality of the coastline using another color or line style. Please add the bottom boarder line of the figure. Is possible to add the evolution of the storm toward the research area for different dates? See the following link as an example. http://www.mwis.org.uk/synoptic-charts. | We plan to replace the existing figure with an alternative which should answer these queries. |

| 12 | Text:     In Section 1, Hm0 increases slightly with the arrival of the storm. | Please clarify this analysis. The exact date of this analysis is not clear. A joint analysis of storm track information and wave parameters must be performed clearly. | We will provide additional information to answer this query. In particular, more information on the storm track and the progression of the storm over time. (This also meets some of the requests in Item (13). |
|---|---|---|---|
| 13 | Text:  unrealistically large wave heights are detected from the 5th February in Section 1 | Please discuss the reason why this behavior (Unrealistically large wave heights) is not observed in the Perranporth location. I suggest including information about the track of the storm and some directional information about the ocean waves. The direction of the swell pattern traveling toward the research area can be very important for wave evolution through time. The evolution of storm direction can help to explain this behavior. | We will expand the discussion to provide the explanation required which will include some extra figures and analysis. |
| 14 | Porthleven experienced the greatest increase in energy content during the first day of the storm, 3rd February. The buoy at Perranporth detected the largest increase in energy content on the second day, 4th | with a low value when compared to Porthleven. Try to discuss the relationship with the buoy location. | We will include some discussion of this based around the differences in wave direction in the two locations and the influence of the nearby coastline. |

| | | | |
|---|---|---|---|
| | February. | | |
| 15 | Text: Perranporth wave buoy is the only buoy located on the northern tip of the south-west. The location explains the observed differences in conditions and is the only location that detected a continued increase in wave height and period on the second day of the storm. The results for Section 1 suggest locations along the northern tip of the south-west coast are more exposed to severe and prolonged wave conditions generated by storms with a track similar to the one in this period. | This is one of the most important topics. For that reason, the quality of the discussion could be increased if a pattern of the spatial wave and storm evolution could be included in the paper. This would help to understand wave transformation toward the research area (other physical processes can be explained better). | It was not the purpose of the paper to provide a spatial analysis of the storm impacts along the coast of Cornwall and Devon, or why two specific locations might experience different wave conditions from one another. The request goes beyond the scope of the paper. However, we propose to add an analysis of wave direction for Section 1 and Section 2 to provide some description of the spatial development of the wave conditions during the storm. |

| 16 | Text: Extensive damage occurred in Section 2 at Dawlish during this storm period, where key railway protection infrastructure along the beach collapsed on the evening of 3 rd and night of 4th February. | Is it possible to include some damage photographs? During this event strong winds were observed, so please consider including some wind patterns (scatterometers could be a good option). | Yes, we will include photographs of the Dawish sea wall damage. We will also include information on winds. (See also response to Item 19). |
|---|---|---|---|
| 17 | Text: in the days leading up to the collapse, high peak wave periods (9 s - 20 s) and swell components (35%-25 45%) are observed. | Please consider putting "were". | Yes, ok. |
| 18 | Text: ibley et al., (2015) suggested that it was the south-southeast wind generated waves in the English Channel | Please consider putting the name of the channel in Figure 1. | We will add 'English Channel' to Figure 1. |

| | | | |
|---|---|---|---|
| 19 | text: A reason for the decrease in Psc could be associated with the arrival of the storm and strong local winds creating locally generated wind waves with high energy. | If the local wind waves are very important, please consider including some information about local wind (magnitude, direction, spatial maps, among others). | See also Item (16). We will include a description of the wind conditions. If space permits we can also add the progression of peak wind speed and direction at various locations along southwest. |
| 20 | Figures 5, 6 and 7 | Please increase the quality of these Figures. As the y axis is too narrow, it is difficult to read the values. Some values look incomplete (cut off at the top of the figure). Discuss the evolution using exact values. | The figure quality will be improved in a revision. |
| 21 | | Have you considered showing the evolution of the unidirectional spectrum by showing the evolution of the sea and swell simultaneously through time?. | This could be shown as an animation of the energy spectrum. We will consider whether it would be possible to construct this to add as online supplementary material from the CCO data. |

| | | | |
|---|---|---|---|
| | | | |
| 22 | Historical comparison | Explain why these months were selected for the analysis. This topic supports the importance of a study area section when the main climatological aspects in the research area must be discussed. A discussion of the annual cycle or stormy season during the year can be helpful. The estimation of the Psc during the month is not clear, please explain better. | We will add an explanation for the choice of months, brief description of the climatological characteristics at the site and further details of the estimation of Psc. |
| 23 | text: Bimodality during the winter period of 2013/2014 was frequent for both Dawlish and Rustington. | Looe bay has similar behavior; please consider including it in the analysis. | We will include a discussion of conditions in Looe Bay. |
| 24 | text: Rustington, located in Section 3 and in the south-east UK could be considered to be | This confirms the importance of using an ocean wave model like SWAN or WWIII to model the spatial | The authors thank for the reviewer for suggesting this, it would indeed be an interesting modelling study. The hypothesis |

| | | | |
|---|---|---|---|
| | less likely to be subjected to swell as its location further east in the English Channel provides greater opportunity for swell waves to be refracted and diffracted towards the shore before penetrating further up the Channel. | behavior of the wind waves during these dates. | about the extent of significant swell penetration up the English Channel has already been made by Mason et al (2008) on the basis of analysing the measurements so we do not feel that including a detailed modelling study here is justified. Our analysis provides additional support to Mason et al's hypothesis and we will include a reference to this in a revision. |
| 25 | text: Bimodality during the winter period of 2013/2014 was frequent for both Dawlish and Rustington | Please try to connect the fact that for Dawlish the highest percentage of bimodality was observed during February 2013/2014 when compared with the other years and the fact that the sea wall collapsed during this month. A technical discussion could be very useful. Why does bimodality increase the failure probability? Looe bay has similar behavior; please consider including it in the analysis. The connection between bimodality and failure probability is not clear. Is there other evidence of bimodality and the failure of other structures? | A discussion along the following lines will be added. Previous laboratory studies (by HR Wallingford) have indicated that the empirical formulae underestimate overtopping if the waves have a bimodal spectrum rather than a unimodal one. This, together with the high percentage of bimodality during the Feb 13/14 storm means that overtopping was likely to be high, and higher than expected by designers of the seawall and flood warning operations managers whose forecasts are based on methods assuming a unimodal wave spectrum. Excessive wave overtopping has been linked to damage to structural damage, (e.g. the |

| | | | EurOtop Guidance), although the link is empirical. Bimodality is mentioned in such guidance but there is little published research on this topic. |
|---|---|---|---|
| 26 | text: As seen in Fig. 9, the mean Psc and the probability of bimodal occurrence is higher at Looe Bay than Dawlish in Section 2 and Rustington in Section 3. The highest mean Psc is observed during the 2013 - 2014 winter storms with high values of Psc throughout December to March (27% -37%) and the highest mean value of Psc over all the years being in February 2014. | This conclusion is not totally clear when comparing Looe Bay and Rustington; please verify. | We have verified this and further clarification will be provided in the revision. |

---

## Author Comment (AC2) · 20 Oct 2018

We thank the reviewers for their time and effort in providing constructive comments on our manuscript. We plan to action most of these (and have supplied counter arguments in cases where no action is proposed) and believe this will result in substantive improvements to the paper. Please find in the supplement our response to the authors. We have compiled this in tabular format for ease of reference and in a single file as many of the comments overlap in either sense or topic.